# The one-loop tadpole in the geoSMEFT

**Tyler Corbett**

Niels Bohr International Academy, Niels Bohr Institute,
University of Copenhagen, Blegdamsvej 17, DK-2100, Copenhagen, Denmark

corbett.t.s@gmail.com

## Abstract

Making use of the geometric formulation of the Standard Model Effective Field Theory we calculate the one-loop tadpole diagrams to all orders in the Standard Model Effective Field Theory power counting. This work represents the first calculation of a one-loop amplitude beyond leading order in the Standard Model Effective Field Theory, and discusses the potential to extend this methodology to perform similar calculations of observables in the near future.

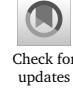 Check for updates

# 1 Introduction

The Standard Model Effective Field Theory (SMEFT) has become a cornerstone of LHC searches for physics beyond the Standard Model (SM). The approach of the SMEFT is to search for the effects of non-resonant heavy new physics, which decouples as $1/\Lambda$, on measurable processes of the known particles. This approach makes two primary assumptions, that the new physics is too heavy to directly produce at a collider and that the Higgs boson belongs to an $SU(2)_L$ doublet, as in the SM. With these assumptions the SMEFT is formulated as a tower of higher-dimensional operators suppressed by the new physics scale $\Lambda$ and added to the SM Lagrangian:

$$\mathcal{L}_{\text{SMEFT}} = \mathcal{L}_{\text{SM}} + \sum_{n=5}^{\infty} \sum_{i} \frac{c_i}{\Lambda^{n-4}} \mathcal{O}_i \,. \tag{1}$$

Each subsequent power of $1/\Lambda$ should therefore be suppressed relative to the last, as $\Lambda$ is a large mass scale well above that of a given scattering process.

For most LHC relevant processes the leading terms come from dimension-six operators suppressed by $\Lambda^2$. There is ongoing discussion on how to handle the truncation of this series in the literature, i.e. to understand the error associated with truncating the series at a given order. Many groups have included squares of dimension-six operator contributions to amplitudes in their work, this allows for an inferred error by comparing results with and without the dimension-six squared term. This presents a theoretical concern – formally this is not the full contribution at order $1/\Lambda^4$ as it neglects dimension-six squared contributions to the amplitude as well as dimension-eight operator effects. There is also the more practical issue, that in many instances the squared term results in more stringent constraints, a result of, for example, chiral suppression of the interference of the $1/\Lambda^2$ term with the SM. This makes a definition of truncation error in this way less than satisfactory.

An alternative approach is to compute the full contribution up to and including $\frac{1}{\Lambda^4}$ effects. This suffers from the seemingly insurmountable number of parameters in the SMEFT beyond leading order. This is to a great degree controlled by only considering resonant processes where four-fermion operators can be neglected as well as making simplifying assumptions on the flavor structure of the SMEFT. To date three works have considered the full $\frac{1}{\Lambda^4}$ dependence in phenomenological studies. In [1], the authors study associated production of a Higgs boson with a $W$ by meticulously elaborating all operators contributing via the Hilbert series method [2–4], and then performing a phenomenological study. Using a similar procedure the authors of [5] study the Drell Yan process at the LHC. In [6], the authors studied $Z$-pole observables and instead used the geometric formulation of the SMEFT which allows for, currently in limited cases, all orders calculations in the SMEFT power counting (i.e. the $1/\Lambda$ power counting).

The geometric SMEFT, or geoSMEFT, was born of an attempt to simplify the one loop calculation of $H \to \gamma\gamma$ [7,8] and the resulting background gauge fixing of the SMEFT [9]. Within this context it was realized that the SMEFT could be formulated in terms of *field-space connection matrices* of the form:

$$M_{I_1 \cdots I_n} \sim \frac{\delta^n \mathcal{L}_{\text{SMEFT}}}{\delta \phi_{I_1} \cdots \delta \phi_{I_n}} \bigg|_{\mathcal{L}(\alpha, \beta, \cdots) \to 0} \,. \tag{2}$$

These field-space connections are then matrices of products of the Higgs doublet with generators of $SU(2)_L$, and the evaluation at $\mathcal{L}(\alpha, \beta, \cdots) \to 0$ represents setting various products of fields and their derivatives to zero. By constructing all gauge-variant, but Lorentz invariant, products of up to any three of the field strengths, covariant derivatives of the scalar field, and products of fermions, the geoSMEFT was formulated to include all three-point functions of SM fields plus arbitrarily many products of scalar fields [10]. This allowed for all-orders (in the

SMEFT power counting) tree-level studies of the SMEFT in [11]. With all three-point functions defined to all orders in the geoSMEFT we can now use an alternative approach to studying the truncation error in the SMEFT. In [6] the full set of $Z$-pole observables at LEP were studied, and an alternative truncation error estimate was proposed - varying the dependence on Wilson coefficients of the $1/\Lambda^4$ result in order to infer the error in the strictly $1/\Lambda^2$ terms.

With an enormous interest being generated around loop calculations in the SMEFT an important next obstacle for the geoSMEFT is to define a similar system for estimating truncation error at one loop. As mentioned above the geoSMEFT only includes vertices of three fields with an arbitrary number of scalar insertions. As such, the geoSMEFT is currently only suitable for the calculation of the tadpole diagram. This article demonstrates the ability to calculate the tadpole at one-loop and all orders in the SMEFT power counting and motivates further development of the geoSMEFT in order to allow consistently defined truncation errors at both tree- and one-loop level.

The article is organized as follows: In Section 2 we define the conventions used in the paper as well as introduce the set of relevant operator forms which contribute to the one-loop tadpole diagram, while in Section 3 we outline the Feynman rules derived from the classical Lagrangian. In Section 4 we gauge fix the geoSMEFT and derive the Feynman rules related to gauge fixing as well as the Feynman rules for ghosts. Then in Section 5 we give the main result of this article, the all orders tadpole, and Sec. 6 is dedicated to discussion of the outlook for the one-loop geoSMEFT and conclusions. The Appendix A includes relevant definitions and relations from the geoSMEFT which are used throughout this article, while App. B demonstrates the importance of the Tadpole diagram both phenomenologically and in preserving the gauge symmetry of the theory beyond tree level.

## 2 Conventions

In order to define the relevant terms of the Lagrangian for the calculation of the tadpole diagram, we follow the formulation of the geoSMEFT given in [10], as well as the gauge fixing of [9] and [12]. We begin by defining the field content of the geoSMEFT, the Higgs doublet of the SM is rewritten in terms of a four-component real scalar field, $\phi^I$, by the following association:

$$H(\phi_I) = \frac{1}{\sqrt{2}} \left[ \begin{array}{c} \phi_2 + i\phi_1 \\ \phi_4 - i\phi_3 \end{array} \right]. \tag{3}$$

The $SU(2)_L$ and $U(1)_Y$ gauge bosons, $B$ and $W^I$, are replaced with four component vector field $W^A = \{W^1, W^2, W^3, B\}$. These weak-eigenstate fields are transformed to the mass basis by the matrices:

$$\mathcal{U}_C^A \equiv \sqrt{g}^{AB} U_{BC}, \qquad \mathcal{V}_K^I \equiv \sqrt{h}^{IJ} V_{JK}. \tag{4}$$

Above and in what follows latin indices are four-component unless otherwise specified. The matrices $\sqrt{g}$ and $\sqrt{h}$ are the inverse-square root expectation value of the field-space connections:[1]

$$
\begin{aligned}
h_{IJ} = {} & \left[ 1 + \phi^2 c_{H\Box}^{(6)} + \sum_{n=0}^{\infty} \left( \frac{\phi^2}{2} \right)^{n+2} (c_{HD}^{(8+2n)} - c_{H,D2}^{(8+2n)}) \right] \delta_{IJ} \\
& + \frac{\Gamma_{A,J}^I \phi_K \Gamma_{A,L}^K \phi^L}{2} \left( \frac{c_{HD}^{(6)}}{2} + \sum_{n=0}^{\infty} \left( \frac{\phi^2}{2} \right)^{n+1} c_{HD,2}^{(8+2n)} \right),
\end{aligned}
\tag{5}
$$

---

[1]Raised indices on field-space connections correspond to inverses of the field-space connection.

$$
\begin{aligned}
g_{AB} &= \left[1 - 4\sum_{n=0}^{\infty}(c_{HW}^{(6+2n)}(1-\delta_{A4}) + c_{HB}^{(6+2n)}\delta_{A4})\left(\frac{\phi^2}{2}\right)^{n+1}\right]\delta_{AB} \\
&\quad - \sum_{n=0}^{\infty}\left(\frac{\phi^2}{2}\right)^n(\phi_I\Gamma_{A,J}^I\phi^J)(\phi_L\Gamma_{B,K}^L\phi^K)(1-\delta_{A4})(1-\delta_{B4}) \\
&\quad + \left[\sum_{n=0}^{\infty}c_{HWB}^{(6+2n)}\left(\frac{\phi^2}{2}\right)^n\right][(\phi_I\Gamma_{A,J}^I\phi^J)(1-\delta_{A4})\delta_{B4} + (A\leftrightarrow B)].
\end{aligned}
\tag{6}
$$

These field space connections correspond to the products of fields: $W_{\mu\nu}^A W^{B,\mu\nu}$ and $(D_\mu\phi)^I(D^\mu\phi)^J$ respectively. As the scalar field $\phi$ is related to its mass eigenstate field $\Phi$ by the inverse square roots of the expectations of these matrices, they are (in the mass eigenstate basis) implicitly dependent on $\sqrt{h}$. The matrices $U$ and $V$ take the weak eigenstate fields and rotate them to the physical basis of the SM, they are given by:

$$
U_{BC} = \begin{bmatrix} \frac{1}{\sqrt{2}} & \frac{1}{\sqrt{2}} & 0 & 0 \\ \frac{i}{\sqrt{2}} & \frac{-i}{\sqrt{2}} & 0 & 0 \\ 0 & 0 & \bar{c}_W & \bar{s}_W \\ 0 & 0 & -\bar{s}_W & \bar{c}_W \end{bmatrix}, \qquad V_{JK} = \begin{bmatrix} \frac{-i}{\sqrt{2}} & \frac{i}{\sqrt{2}} & 0 & 0 \\ \frac{1}{\sqrt{2}} & \frac{1}{\sqrt{2}} & 0 & 0 \\ 0 & 0 & -1 & 0 \\ 0 & 0 & 0 & 1 \end{bmatrix}.
\tag{7}
$$

$\mathcal{U}$ and $\mathcal{V}$ transform the weak eigenstate basis fields, $W$ and $\phi$, to the physical basis fields $A^B = \{W^+, W^-, Z, A\}$ and $\Phi^I = \{\Phi^-, \Phi^+, \chi, h\}$. Note, $h$ is used to denote the Higgs boson as well as the field-space connection of Eq. 5. When the $h$ field-space connection is used it has either indices or appears as $\sqrt{h}$. According to the above, the bosonic fields are rotated to the mass basis as:

$$
A^B = \mathcal{U}_C^B W^C, \qquad\qquad \Phi^I = \mathcal{V}_K^I \phi^K.
\tag{8}
$$

The barred Weinberg angles, $\bar{s}_W$ and $\bar{c}_W$ are defined in the Appendix. In addition to the above we also have the ghosts for the electroweak gauge bosons, $u^{\mathcal{A}} = \mathcal{U}_C^{\mathcal{A}} u^C$, the gluon field $G^{\mathcal{A}}$ and the corresponding ghost $u_G^{\mathcal{A}}$. The gluons and their corresponding ghosts are transformed to canonically normalized fields by:

$$
G^{\mathcal{A}} = \sqrt{\kappa^{-1}}\mathcal{G}^{\mathcal{A}}, \qquad\qquad u_G^{\mathcal{A}} = \sqrt{\kappa^{-1}}u_{\mathcal{G}}^{\mathcal{A}},
\tag{9}
$$

$\kappa$ is defined below, and is the field space connection of the combination $\mathcal{G}_{\mu\nu}^{\mathcal{A}}\mathcal{G}^{\mathcal{A},\mu\nu}$. Script latin indices are $SU(3)_c$ gluon indices. $G$ corresponds to the canonically normalized gluonic field, while $\mathcal{G}$ corresponds to the gluonic field before the kinetic term is transformed. In this article, fermionic fields only occur in loops and are therefore always summed over, as such we use the short hand $\psi$ for all fermionic fields.

The full set of operator forms contributing to two- and three-point functions of the SMEFT was derived in [10]. They include:

$$
\begin{aligned}
&h_{IJ}(D_\mu\phi)^I(D_\mu\phi)^J, &&g_{AB}W_{\mu\nu}^A W^{B\mu\nu}, &&\kappa_{IJ}^A(D_\mu\phi)^I(D_\nu\phi)^J W_A^{\mu\nu}, \\
&\mathcal{Y}^\psi\bar{\psi}_1\psi_2, &&\kappa\mathcal{G}_{\mu\nu}^{\mathcal{A}}\mathcal{G}^{\mathcal{A}\mu\nu}, && \\
&f_{ABC}W_{\mu\nu}^A W^{B,\nu\rho}W_\rho^{C,\mu}, &&d_A\bar{\psi}_1\sigma^{\mu\nu}\psi_2\mathcal{W}_{\mu\nu}^A, &&\kappa_{\mathcal{ABC}}\mathcal{G}_{\mu\nu}^{\mathcal{A}}\mathcal{G}^{\mathcal{B},\nu\rho}\mathcal{G}_\rho^{\mathcal{C},\mu}, \\
&c\bar{\psi}_1\sigma^{\mu\nu}T_{\mathcal{A}}\psi_2\mathcal{G}_{\mu\nu}^{\mathcal{A}}, &&L_{IA}\bar{\psi}_1\gamma^\mu\sigma_A\psi_2(D_\mu\phi)^I.
\end{aligned}
\tag{10}
$$

The covariant derivative of the four component scalar and the field strength tensors of the

vectors are then defined as:

$$(D_\mu \phi)^I = \left( \partial^\mu \delta^I_J - \frac{1}{2} W^{A,\mu} \tilde{\gamma}^I_{A,J} \right) \phi^J, \tag{11}$$

$$W^A_{\mu\nu} = \partial_\mu W^A_\nu - \partial_\nu W^A_\mu - \tilde{\epsilon}^A_{\ BC} W^B_\mu W^C_\nu, \tag{12}$$

$$\mathcal{G}^A_{\mu\nu} = \partial_\mu \mathcal{G}^A_\nu - \partial_\nu \mathcal{G}^A_\mu - f^A_{\ BC} \mathcal{G}^B_\mu \mathcal{G}^C_\nu. \tag{13}$$

The matrices $\tilde{\gamma}^I_{A,J}$ and $\tilde{\epsilon}^A_{\ BC}$ are defined in the Appendix. The $f^A_{\ BC}$ are the usual structure constants of $SU(3)_c$.

In addition to the operators defined in Eq. 10 we also define the all-orders Higgs potential,

$$V(\phi^I) = \frac{\lambda}{4} \left( \phi^2 - v_0^2 \right)^2 - \sum_{n=1}^{\infty} c_H^{(4+2n)} \left( \frac{\phi^2}{2} \right)^{2+n}. \tag{14}$$

In the above, $v_0$ is the vacuum expectation value that minimizes the tree-level Higgs potential for the SM. Spontaneous symmetry breaking occurs in the geoSMEFT for $\phi^I \to v\delta^{I4} + \sqrt{h}^{IJ} V_{JK} \Phi^K$,[2] where $v$ is the vacuum expectation value which minimizes the tree level potential of the geoSMEFT. $c_H^{(4+2n)}$ is the Wilson coefficient of the dimension $4+2n$ pure Higgs operator suppressed by the heavy mass scale $\Lambda^{2n}$, this $\Lambda$ dependence is absorbed into the Wilson coefficient here and for the operators below for convenience. At tree level, requiring the coefficient of the tadpole term in the potential be zero gives the relation between $v_0$ and $v$:

$$t = 0 \propto v^2 - \frac{1}{\lambda} \sum_{n=1}^{\infty} \frac{(4+2n)v^{2+2n}}{2^{2+n}} c_H^{(2n+4)} - v_0^2. \tag{15}$$

We note that solving this equation for $v^2$ requires numerical methods for $n \geq 4$ as it is a polynomial of order $n+1$ in $v^2$.

In what follows we will derive the one-loop correction to this result to all orders in the SMEFT power counting. The choice of $t = 0$ at one loop corresponds to the FJ tadpole scheme [13], with this choice we choose to expand about the true (one-loop) vacuum. This simplifying choice means tadpole diagrams need not be included in one-loop calculations (the tadpole and its counter term exactly cancel), however the loop improved vacuum expectation value needs to be used in tree level calculations. Further, this one-loop result is required to demonstrate the gauge invariance of observables, such as the masses of the gauge bosons in the on-shell renormalization scheme [14,15]. This is discussed in Appendix B as well as in the conclusions.

The terms from Eq. 10 which contribute to the one-loop tadpole diagram are those which involve a single Higgs boson coupling to two fermions, gauge bosons, or additional scalars. As such the last two lines do not contribute as they include three or more particles other than the Higgs boson and therefore only contribute at higher loop order. In the case of the connection $L_{IA}$ there is no contribution as these operators correspond to the Hermitian derivative form, $(H^\dagger \overset{\leftrightarrow}{D}_\mu H)(\bar{\psi}\gamma^\mu \psi)$, which causes the Higgs-fermion couplings to vanish identically. While the operators coupling the Higgs boson to gluons will result in scale-less loop integrals which vanish identically, we retain them as the all-orders Feynman rules derived from the $\kappa_{AB}$ operator form are the simplest and serve as intuitive examples of how the rules are derived. Reproducing the all-orders form of the relevant connections from [10] we have (in addition

---

[2]This is a convenient choice of how to realize spontaneous symmetry breaking in the geoSMEFT which is consistent with $\langle H^\dagger H \rangle = v^2/2$ [12].

to Eqs. 5 and 6 above):

$$
\begin{aligned}
\kappa_{IJ}^{A} &= -\frac{1}{2}\gamma_{4,J}^{I}\delta_{A4}\sum_{n=0}^{\infty}c_{HDHB}^{(8+2n)}\left(\frac{\phi^2}{2}\right)^{n+1} - \frac{1}{2}\gamma_{A,J}^{I}(1-\delta_{A4})\sum_{n=0}^{\infty}c_{HDHW}^{(8+2n)}\left(\frac{\phi^2}{2}\right)^{n+1} \\
&\quad -\frac{1}{8}(1-\delta_{A4})[\phi_K\Gamma_{A,L}^{K}\phi^L][\phi_M\Gamma_{B,L}^{M}\phi^N]\gamma_{B,J}^{I}\sum_{n=0}^{\infty}c_{HDHW,3}^{(10+2n)}\left(\frac{\phi^2}{2}\right)^{n} \\
&\quad +\frac{1}{4}\epsilon_{ABC}[\phi_K\Gamma_{B,L}^{K}\phi^L]\gamma_{C,J}^{I}\sum_{n=0}^{\infty}c_{HDHW,2}^{(8+2n)}\left(\frac{\phi^2}{2}\right)^{n},
\end{aligned}
\tag{16}
$$

$$
\mathcal{Y}_{pr}^{\psi} = -\overset{(\sim)}{H}(\phi_I)[Y_\psi]^\dagger + \overset{(\sim)}{H}(\phi_I)\sum_{n=0}^{\infty}c_{\psi H,pr}^{(6+2n)}\left(\frac{\phi^2}{2}\right)^{n+1},
\tag{17}
$$

$$
\kappa = \left[1-4\sum_{n=0}^{\infty}c_{HG}^{(6+2n)}\left(\frac{\phi^2}{2}\right)^{n+1}\right].
\tag{18}
$$

Where $\overset{(\sim)}{H}$ is the Higgs doublet for leptons and down quarks, and $\epsilon_{ij}H^j$ for up quarks. The matrices $\Gamma_{A,J}^{I}$ and $\gamma_{A,J}^{I}$ are given in the Appendix for brevity. We have also used $\phi^2 = \phi^I\phi_I = \phi_I\delta^{IJ}\phi_J$. The $c_i^{(n)}$ are the Wilson coefficients of operators of dimension $n$ and are suppressed by a factor of $\Lambda^{n-4}$ which has been absorbed into their definition for the sake of compactness of these and the following expressions. The inverse-square root of $g_{IJ}$ and $h_{IJ}$ are the matrices of Eq. 4 which, with the matrices $U$ and $V$, take the weak eigenstate fields to the mass eigenstate fields of the SMEFT. Latin indices $A, B, \cdots$ are those associated with the four-component representation of the gauge boson indices for $SU(2)_L \times U(1)_Y$, $I, J \cdots$ are are the four-component indices associated with the four-component real scalar field, and $\mathcal{A}, \mathcal{B}$ are associated with color indices of the gluons. Fermionic indices are generally suppressed.

The above is all that is needed to define the relevant all-orders three-point functions for the classical Lagrangian in the geoSMEFT:

$$
\begin{aligned}
\mathcal{L}_{cl}(\phi^I, W^A, \mathcal{G}^{\mathcal{A}}, \psi) &= h_{IJ}(D_\mu\phi)^I(D_\mu\phi)^J - V(\phi) + g_{AB}W_{\mu\nu}^{A}W^{B,\mu\nu} + \kappa\mathcal{G}_{\mu\nu}^{\mathcal{A}}\mathcal{G}^{\mathcal{A},\mu\nu} \\
&\quad + \kappa_{IJ}^{A}(D_\mu\phi)^I(D_\nu\phi)^J W_A^{\mu\nu} + \sum_\psi \mathcal{Y}\bar{\psi}_1\psi_2.
\end{aligned}
\tag{19}
$$

In Section 4 we will choose to adopt the background field method of gauge fixing. Therefore in the discussion of the classical Lagrangian that follows we will double the bosonic field content of the above Lagrangian as:

$$
\mathcal{L}_{cl}(\phi^I, W_\mu^A, \mathcal{G}_\mu^{\mathcal{A}}, \psi) \to \mathcal{L}_{cl}(\phi^I + \hat{\phi}^I, W^A + \hat{W}^A, \mathcal{G}^{\mathcal{A}} + \hat{\mathcal{G}}^{\mathcal{A}}, \psi).
\tag{20}
$$

Where the hatted fields are referred to as the background fields and the unhatted as the quantum fields. The choice of the background field method has various advantages, one of which is the preservation of the naive Ward Identities as discussed in [12,16,17]. This methodology has been adopted in many SMEFT related publications because of its nice properties, see for example [7,18,19]. In this methodology the quantum fields are gauge fixed, while the background fields are not. As fermionic fields are not involved in the gauge fixing they are not split into background and quantum fields. As such all external particles for a given amplitude correspond to background field while internal lines are quantum fields. Therefore in what follows we derive the couplings of the background Higgs boson field, $\hat{h}$, to two quantum fields.

# 3 The all-orders vertices

In order to define the relevant three-point functions for the one-loop tadpole diagrams we must obtain the relevant Feynman rules from Eq. 10. We will do this while preserving the form of the field-space connections when possible in order to maintain results that are manifestly all orders in the $\frac{1}{\Lambda^n}$ power counting. The Feynman rules that follow were checked using `FeynRules`. They can be understood as follows: the subscript of a field in {} corresponds to the momenta, Lorentz indices, and color indices with the same subscript on the right side of the equations below. In the case of a field with no subscript, the Feynman rule does not depend explicitly on that field's properties.

The simplest Feynman rules to derive are from the field space connections $g_{AB}$, $\kappa_{AB}$, and $Y_{pr}^{\psi}$ as the Higgs dependence is purely in the connection matrix. Varying Eq. 18 with respect to the background field $\hat{h}$ gives the coupling of a Higgs boson to two gluons:

$$\{\hat{h}, G_1, G_2\} \quad = \quad i \left\langle \frac{\delta \kappa}{\delta \hat{h}} \right\rangle \left( \sqrt{\kappa^{-1}} \right)^2 \Pi_{1,2} \delta^{\mathcal{A}_1 \mathcal{A}_2} . \tag{21}$$

Where, for convenience, we have defined,

$$\Pi_{1,2} \equiv (p_1^{\mu_2} p_2^{\mu_1} - p_1 \cdot p_2 \eta^{\mu_1 \mu_2}) . \tag{22}$$

It should be noted there are implied rotations of the quantity $\phi_I$ within the field-space connections such as $\kappa$: beyond leading order $\sqrt{\kappa}$ is a function of $\phi^I = \sqrt{h}^{IJ} V_{JK} \Phi^K$. Explicitly taking the variations gives instead:

$$\{\hat{h}, G_1, G_2\} \quad \rightarrow \quad i \sqrt{h}^{44} \left( \sqrt{\kappa^{-1}} \right)^2 v_T \sum_{i=0}^{\infty} \frac{v_T^{2n}(n+1)}{2^{n-2}} c_{HG}^{(6+2n)} \Pi_{1,2} \delta^{\mathcal{A}_1 \mathcal{A}_2} . \tag{23}$$

Similarly for the yukawa-like couplings:

$$\{\hat{h}, \bar{\psi}_r, \psi_r\} \quad = \quad -i \left\langle \frac{\delta \mathcal{Y}_{rr}^{\psi}}{\delta \hat{h}} \right\rangle = i \frac{\sqrt{h}^{44}}{v} \bar{M}_{\psi,rr} - i \frac{\sqrt{h}^{44}}{\sqrt{2}} \sum_{n=0}^{\infty} c_{\psi H,rr}^{(6+2n)} \frac{v^{2n+2}}{2^{n+1}} (2n+2) . \tag{24}$$

As only like-flavors will contribute to the Tadpole diagram we have only considered diagonal entries of $\mathcal{Y}^{\psi}$ and substituted in terms of the barred tree-level masses of the fermions. The tree-level fermion mass to all orders is simply the expectation of the field connection $\mathcal{Y}$ of Eq. 17:

$$\bar{M}_{\psi} = \langle (\mathcal{Y}^{\psi})^{\dagger} \rangle . \tag{25}$$

The remaining terms are more complicated than the above, as such we only write the vertex functions in terms of variations on the field-space connections. Some examples of the field-space connections expanded in terms of Wilson coefficients can be found in the Appendix. The coupling to two gauge bosons coming from $g_{AB}$ is given by:

$$\{\hat{h}, W_1^+, W_2^-\} \quad = \quad -i \left\langle \frac{\delta g_{11}}{\delta \hat{h}} \right\rangle (\sqrt{g}^{11})^2 \Pi_{1,2} , \tag{26}$$

$$\{\hat{h}, A_1, A_2\} \quad = \quad -i \Sigma_{AA} \Pi_{1,2} , \tag{27}$$

$$\{\hat{h}, Z_1, Z_2\} \quad = \quad -i \Sigma_{ZZ} \Pi_{1,2} , \tag{28}$$

$$\tag{29}$$

$$\Sigma_{AA} \equiv \sum_{A,B=3}^{4} \left( \bar{c}_W^2 \left\langle \frac{\delta g_{AB}}{\delta \hat{h}} \right\rangle \sqrt{g}^{A4} \sqrt{g}^{B4} + 2\bar{c}_W \bar{s}_W \left\langle \frac{\delta g_{AB}}{\delta \hat{h}} \right\rangle \sqrt{g}^{3A} \sqrt{g}^{B4} + \bar{s}_W^2 \left\langle \frac{\delta g_{AB}}{\delta \hat{h}} \right\rangle \sqrt{g}^{3A} \sqrt{g}^{3B} \right), \tag{30}$$

$$\Sigma_{ZZ} \equiv \sum_{A,B=3}^{4} \left( \bar{c}_W^2 \left\langle \frac{\delta g_{AB}}{\delta \hat{h}} \right\rangle \sqrt{g}^{3A} \sqrt{g}^{3B} - 2\bar{c}_W \bar{s}_W \left\langle \frac{\delta g_{AB}}{\delta \hat{h}} \right\rangle \sqrt{g}^{3A} \sqrt{g}^{B4} + \bar{s}_W^2 \left\langle \frac{\delta g_{AB}}{\delta \hat{h}} \right\rangle \sqrt{g}^{A4} \sqrt{g}^{B4} \right)$$

$$= \Sigma_{AA}(\bar{s}_W \rightarrow -\bar{c}_W, \bar{c}_W \rightarrow \bar{s}_W). \tag{31}$$

In order to form a tadpole diagram from the connection $\kappa_{IJ}^A$ one of the covariant derivatives must generate a vector boson while the other must correspond to the background Higgs boson, as such the rules are straightforward to derive as well:

$$\{\hat{h}_1, W_2^+, W_3^-\} = \bar{g}_2 \sqrt{g}^{11} \sqrt{h}^{44} v \left[ (\langle \kappa_{13}^1 \rangle - i\langle \kappa_{14}^1 \rangle) p_1^{\mu_2} p_2^{\mu_3} - (\langle \kappa_{13}^1 \rangle + i\langle \kappa_{14}^1 \rangle) p_1^{\mu_3} p_3^{\mu_2} \right.$$
$$\left. + \left( \langle \kappa_{13}^1 \rangle [p_1 \cdot p_3 - p_1 \cdot p_2] + i\langle \kappa_{14}^1 \rangle [p_1 \cdot p_2 + p_1 \cdot p_3] \right) \eta^{\mu_2 \mu_3} \right], \tag{32}$$

$$\{\hat{h}_1, Z_2, Z_3\} = -i\sqrt{h}^{44} \bar{g}_Z v \left[ \left( \bar{c}_W \sqrt{g}^{33} - \bar{s}_W \sqrt{g}^{34} \right) \langle \kappa_{34}^3 \rangle + \left( \bar{s}_W \sqrt{g}^{44} - \bar{c}_W \sqrt{g}^{34} \right) \langle \kappa_{12}^4 \rangle \right]$$
$$\times \left[ p_1^{\mu_2} p_2^{\mu_3} + p_1^{\mu_3} p_3^{\mu_2} - p_1 \cdot (p_2 + p_3) \eta^{\mu_2 \mu_3} \right]. \tag{33}$$

No coupling to the photon is generated as one of the vector bosons must come from the covariant derivative which has no $A$ dependence for the Higgs boson. In simplifying these expressions we have used:

$$\langle \kappa_{13}^1 \rangle = -\langle \kappa_{24}^1 \rangle = -\langle \kappa_{31}^1 \rangle = \langle \kappa_{42}^1 \rangle = \langle \kappa_{14}^2 \rangle = \langle \kappa_{23}^2 \rangle = -\langle \kappa_{32}^2 \rangle = -\langle \kappa_{41}^2 \rangle, \tag{34}$$

$$\langle \kappa_{14}^1 \rangle = \langle \kappa_{23}^1 \rangle = -\langle \kappa_{32}^1 \rangle = -\langle \kappa_{41}^1 \rangle = -\langle \kappa_{13}^2 \rangle = \langle \kappa_{24}^2 \rangle = \langle \kappa_{31}^2 \rangle = -\langle \kappa_{42}^2 \rangle, \tag{35}$$

$$\langle \kappa_{12}^4 \rangle = -\langle \kappa_{34}^4 \rangle. \tag{36}$$

In addition to the fact $\kappa_{IJ}^A$ is antisymmetric. As the rules for interactions derived from $\kappa_{IJ}^A$ necessarily depend on the momentum of the background Higgs boson (i.e. one of the derivatives must be acting on the Higgs boson) these rules will not contribute to the tadpole diagram.

Finally, the Feynman rules arising from the field-space connection $h_{IJ}$ are slightly more complicated as the background Higgs boson can come from either the metric or the $(D_\mu \phi)$ terms. These operator forms also contribute not only to Higgs-gauge couplings, but also to Higgs-goldstone couplings. For $\hat{h}$ sourced from the field space connection we have the following rules:

$$\{\hat{h}, \Phi_1^0, \Phi_1^0\} = -i\left\langle \frac{\delta h_{33}}{\delta \hat{h}} \right\rangle (\sqrt{h}^{33})^2 p_1 \cdot p_2, \tag{37}$$

$$\{\hat{h}, \Phi_1^+, \Phi_2^-\} = -i\left\langle \frac{\delta h_{11}}{\delta \hat{h}} \right\rangle (\sqrt{h}^{11})^2 p_1 \cdot p_2, \tag{38}$$

$$\{\hat{h}, h_1, h_2\} = -i\left\langle \frac{\delta h_{44}}{\delta \hat{h}} \right\rangle (\sqrt{h}^{44})^2 p_1 \cdot p_2, \tag{39}$$

$$\{\hat{h}, W_1^+, W_2^-\} = i\left\langle \frac{\delta h_{11}}{\delta \hat{h}} \right\rangle \bar{M}_W^2 (\sqrt{h}^{11})^2 \eta_{\mu_1 \mu_2}, \tag{40}$$

$$\{\hat{h}, Z_1, Z_2\} = i\left\langle \frac{\delta h_{33}}{\delta \hat{h}} \right\rangle \bar{M}_Z^2 (\sqrt{h}^{33})^2 \eta_{\mu_1 \mu_2}. \tag{41}$$

The coupling $\hat{h}\gamma\gamma$ vanishes identically, which follows from the fact the operator forms of the field-space connection $h_{IJ}$ correspond to rescalings of the SM Higgs couplings to vector bosons.

In the case that $\hat{h}$ is sourced from the covariant derivative terms we have two contributions. The first is from the $\langle h \rangle$ which can only generate $\hat{h}$-vector three point functions:[3]

$$\{\hat{h}, W_1^+, W_2^-\} = 2i\sqrt{h}^{44}\frac{\bar{M}_W^2}{v}\eta_{\mu_1\mu_2}, \tag{42}$$

$$\{\hat{h}, Z_1, Z_2\} = 2i\sqrt{h}^{44}\frac{\bar{M}_Z^2}{v}\eta_{\mu_1\mu_2}. \tag{43}$$

As above, the $\hat{h}\gamma\gamma$ coupling vanishes identically. Secondly, $\hat{h}$ couplings to goldstone bosons from variations of the metric with respect to the goldstone bosons could be present, however they vanish identically.

In addition to the above we need to include terms like $c_H^{(2n-4)}(H^\dagger H)^{2n}$. The Feynman rules for $\hat{h}$ coupling to two quantum fields can be generalized from Eq. 4.2 of [10] by using the multinomial coefficient:

$$\{\hat{h}, h, h\} = -2i(\sqrt{h}^{44})^3 v\left[3\lambda - \sum_{n=3}^{\infty}\frac{1}{2^n}\binom{2n}{1,2,2n-3}v^{2n-4}c_H^{(2n)}\right], \tag{44}$$

$$\{\hat{h}, \Phi^0, \Phi^0\} = -2i(\sqrt{h}^{33})^2\sqrt{h}^{44} v\left[\lambda - \sum_{n=3}^{\infty}\frac{1}{2^{n-1}}\binom{n}{1,1,n-2}v^{2n-4}c_H^{(2n)}\right], \tag{45}$$

$$\{\hat{h}, \Phi^+, \Phi^-\} = -i(\sqrt{h}^{11})^2\sqrt{h}^{44} v\left[2\lambda - \sum_{n=3}^{\infty}\frac{1}{2^{n-2}}\binom{n}{1,1,n-2}v^{2n-4}c_H^{(2n)}\right]. \tag{46}$$

In the above the multinomial for $\hat{h}h^2$ can be understood to come from $(v+\hat{h}+h)^{2n}$ terms, the $\Phi^0$ rule from $[(\Phi^0)^2 + 2\hat{h}v + v^2]^n$, and the rule for $\Phi^\pm$ from $[2|\Phi^+|^2 + 2\hat{h}v + v^2]^n$. This explains the minor differences between the Feynman rules above.

The above constitute all the rules from the classical Lagrangian necessary to perform the calculation of the tadpole diagrams to all orders in the SMEFT power counting, what remains are the gauge-fixing and ghost contributions.

## 4  Gauge fixing the geoSMEFT

Background gauge fixing for the SMEFT was performed first in [9]. This was first done for the gluons in [18], then later repeated in [16] in a manner more consistent with the gauge fixing of the weak gauge bosons of [9] which is adopted here. The gauge fixing terms are given by:

$$\mathcal{L}_{GF} = -\frac{\hat{g}_{AB}}{2\xi_W}\mathcal{G}^A\mathcal{G}^B - \frac{\kappa}{2\xi_G}\mathcal{G}^{\mathcal{A}}_{\text{color}}\mathcal{G}^{\mathcal{A}}_{\text{color}}, \tag{47}$$

$$\mathcal{G}^A = \partial_\mu W^{A,\mu} - \tilde{\epsilon}^A_{\ BC}\hat{W}^B_\mu W^{C\mu} + \frac{\xi}{2}\hat{g}^{AB}\phi^I\hat{h}_{IK}\tilde{\gamma}^K_{B,J}\hat{\phi}^J, \tag{48}$$

$$\mathcal{G}^{\mathcal{A}}_{\text{color}} = \partial_\mu G^{\mu,\mathcal{A}} - g_3 f^{\mathcal{ABC}}\hat{G}_{\mu,\mathcal{B}}G^\mu_{\mathcal{C}}. \tag{49}$$

Where in the above, unhatted fields are understood to be quantum fields and the hatted field-space connections are the normal field space connections (i.e. $\hat{g}$ and $\hat{h}$) with all quantum fields set to zero. This notational choice is also the case below in the ghost Lagrangian. Starting with the gluonic gauge fixing as it is the simplest we obtain the Feynman rule:

$$\{\hat{h}, G_1, G_2\} = \frac{i}{\xi_G}\left\langle\frac{\delta\kappa}{\delta\hat{h}}\right\rangle(\sqrt{\kappa^{-1}})^2 p_1^{\mu_1}p_2^{\mu_2}\delta^{\mathcal{A}_1\mathcal{A}_2}. \tag{50}$$

---

[3]Also $\hat{h}\Phi^{0,\pm}$-vector couplings which do not contribute to the Tadpole diagram.

In the case of the electroweak gauge fixing a coupling of the background Higgs field to gauge bosons can be obtained from the variation with respect to the field-space connection of Eq. 47 and the square of the derivative term of Eq. 48. The second terms of Eqs. 48 and 49 cannot contribute as they include a background gauge field, while the final term allows for a $\hat{h}$ coupling to goldstone bosons when all but one of the $\hat{g}$, $\hat{h}$, and $\hat{\phi}$ are set to their expectation values. This results in the following Feynman Rules:

$$\{\hat{h}, W_1^+, W_2^-\} = \frac{i}{\xi_W} \left\langle \frac{\delta g_{11}}{\delta \hat{h}} \right\rangle (\sqrt{g}^{11})^2 p_1^{\mu_1} p_2^{\mu_2}, \tag{51}$$

$$\{\hat{h}, A_1, A_2\} = \frac{i}{\xi_W} \Sigma_{AA} p_1^{\mu_1} p_2^{\mu_2}, \tag{52}$$

$$\{\hat{h}, Z_1, Z_2\} = \frac{i}{\xi_W} \Sigma_{ZZ} p_1^{\mu_1} p_2^{\mu_2}, \tag{53}$$

$$\{\hat{h}, \Phi^+, \Phi^-\} = -i \frac{\bar{M}_W^2}{v} \left[ 2 \left\langle \frac{\delta h_{11}}{\delta \hat{h}} \right\rangle (\sqrt{h}^{11})^2 v + 2\sqrt{h}^{44} + \left\langle \frac{\delta g^{11}}{\delta \hat{h}} \right\rangle (\sqrt{g}_{11})^2 v \right] \xi_W, \tag{54}$$

$$\{\hat{h}, \Phi^0, \Phi^0\} = -i \frac{\bar{M}_Z^2}{v} \left[ 2 \left\langle \frac{\delta h_{33}}{\delta \hat{h}} \right\rangle (\sqrt{h}^{33})^2 v + 2\sqrt{h}^{44} - \Sigma_{ZZ} v \right] \xi_W. \tag{55}$$

Note no $\hat{h}$ coupling to two quantum Higgs bosons is generated.

The ghost Lagrangian was also derived in [9],[4] it is reproduced here excluding any terms with gauge fields as they cannot contribute to the one-loop Tadpole diagram (the ghost Lagrangian is by definition quadratic in the ghost fields):

$$\mathcal{L}_{\text{ghost}} = -\hat{g}_{AB} \bar{u}^B \left[ \partial^2 + \frac{\xi_W}{4} \hat{g}^{AD} (\phi^J + \hat{\phi}^J) \tilde{\gamma}_{CJ}^I \hat{h}_{IK} \tilde{\gamma}_{DL}^K \hat{\phi}^L \right] u^C - \hat{\kappa} \bar{u}_\mathcal{A}^G \partial^2 u_\mathcal{A}^G. \tag{56}$$

As was the case for the gauge fixing terms, $\hat{h}\bar{u}u$ couplings can be obtained either from a variation with respect to one of the field-space connections or explicitly from $\hat{\phi}$, $\hat{h}$, or $\hat{g}$:

$$\{\hat{h}, \bar{u}_1^G, u_2^G\} = i \left\langle \frac{\delta \kappa}{\delta \hat{h}} \right\rangle (\sqrt{\kappa^{-1}})^2 p_2^2 \delta_{\mathcal{A}_1 \mathcal{A}_2}, \tag{57}$$

$$\{\hat{h}, \bar{u}_1^{W^+}, u_2^{W^+}\} = -i \left[ \left\langle \frac{\delta h_{11}}{\delta \hat{h}} \right\rangle \bar{M}_W^2 (\sqrt{h}^{11})^2 \xi + 2\bar{M}_W^2 \sqrt{h}^{44} \xi - (\sqrt{g}^{11})^2 \left\langle \frac{\delta g_{11}}{\delta \hat{h}} \right\rangle p_2^2 \right]$$

$$= \{\hat{h}, \bar{u}_1^{W^-}, u_2^{W^-}\}, \tag{58}$$

$$\{\hat{h}, \bar{u}_1^\gamma, u_2^\gamma\} = i \Sigma_{AA} p_2^2, \tag{59}$$

$$\{\hat{h}, \bar{u}_1^Z, u_2^Z\} = i \Sigma_{ZZ} p_2^2 - i \bar{M}_Z^2 \left( 2\sqrt{h}^{44} + (\sqrt{h}^{33})^2 \left\langle \frac{\delta h_{33}}{\delta \hat{h}} \right\rangle \right) \xi. \tag{60}$$

In the case of the ghosts associated with the photon, the $\xi$ dependent term vanishes identically. This is analogous to the case of the classical contribution from the field space metric $h_{IJ}$, see the discussions around Eqs. 41 and 43. Note that in the case of the ghost for the photon field we have used the notation $u^\gamma$ to distinguish the field from the four-component ghost field $u^A$. With the above, all Feynman rules necessary to calculate the tadpole diagram at one loop and to all orders in the SMEFT expansion are now defined.

## 5 The all-orders SMEFT tadpole

The one loop diagrams that contribute are shown in Figure 1, as was noted in Section 2 the Feynman rules coupling the Higgs boson to gluons as well as those coupling the Higgs boson

---

[4]Here we have adopted the sign choice of [18].

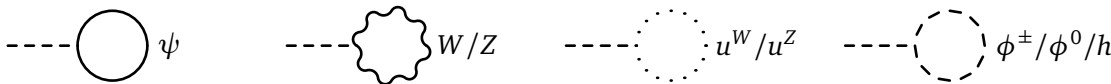

Figure 1: One loop diagrams contributing to the Tadpole. The photon and gluons and their corresponding ghosts do not contribute as they are massless the loop integrals are identically zero.

to colored ghosts do not contribute to the tadpole diagram as the loop integral is scaleless. Making use of dimensional regularization in $d = 4 - 2\epsilon$ dimensions, the fermionic couplings result in the following contribution at one loop:

$$
T_H^\psi = -\frac{N_c \bar{M}_\psi}{4\pi^2} \left\langle \frac{\delta \mathcal{Y}^\psi}{\delta \hat{h}} \right\rangle A_0(\bar{M}_\psi) \tag{61}
$$

$$
= \frac{N_c \bar{M}_\psi}{4\pi^2} \sqrt{h}^{44} \left( \frac{\bar{M}_\psi}{v} - \frac{1}{\sqrt{2}} \sum_{n=0}^{\infty} \frac{v^{2n+2}}{2^{n+1}} (2n+2) c_{\psi H}^{(n)} \right) A_0(\bar{M}_\psi) \tag{62}
$$

$$
= \frac{N_c \bar{M}_\psi}{4\pi^2} \left[ \frac{\bar{M}_\psi}{v} - \frac{v}{4} \left( 2\sqrt{2} v c_{\psi H}^{(6)} + \bar{M}_\psi [c_{HD}^{(6)} - 4c_{H\Box}^{(6)}] \right) - \frac{v^4}{8} \sqrt{2} \left( c_{\psi H}^{(8)} + [4c_{H\Box}^{(6)} - c_{HD}^{(6)}] c_{\psi H}^{(6)} \right) \right.
$$
$$
\left. + \frac{\bar{M}_\psi}{32} \left( 4c_{HD}^{(8)} + 4c_{HD,2}^{(8)} - 3[c_{HD}^{(6)} - 4c_{H\Box}^{(6)}]^2 \right) \right] A_0(\bar{M}_\psi) + \mathcal{O}\left( \frac{1}{\Lambda^6} \right). \tag{63}
$$

Where we have used the Passarino-Veltman scalar A function,

$$
A_0(M) = M^2 \left[ 1 + \frac{1}{\epsilon} - \gamma_E + \log\left( \frac{4\pi \mu^2}{M^2} \right) \right]. \tag{64}
$$

The three equivalences of Eq. 63 show first the geoSMEFT result, the result with the variation of the field-space connection written explicitly in terms of the relevant Wilson coefficients while keeping the compact form for the transformations that canonically normalizes the Higgs background field, and finally the full expansion in terms of the Wilson coefficients to order $\frac{1}{\Lambda^4}$. The barred quantities are not expanded as they are more closely related to input parameters that would be chosen in a phenomenological study, this also serves to simplify the expressions so they fit in paper format. This demonstrates that the geoSMEFT trivially sums the Wilson coefficient dependence of the SMEFT. In a traditional SMEFT approach one would enumerate all the contributing operators to a given order in the SMEFT power counting and the corresponding Feynman rules, perform the calculations, and again expand to a given order. Here we perform the all orders calculation and can expand to a given order after the full calculation is performed.

The compactness of the expressions also allows for a cleaner understanding of cancellations in the theory such as in the case of cancellations between gauge-boson, ghost, and goldstone boson contributions as we see next. Below we neglect to expand in terms of individual Wilson coefficients until the terms are added together as many simplifications occur after summing the diagrams. In the case of the $W$ and $Z$ bosons we have:

$$
T_H^W = \frac{\bar{M}_W^2}{16\pi^2} \left[ (\sqrt{g}^{11})^2 \left\langle \frac{\delta g_{11}}{\delta \hat{h}} \right\rangle - \frac{2}{v} \sqrt{h}^{44} - \left\langle \frac{\delta h_{11}}{\delta \hat{h}} \right\rangle (\sqrt{h}^{11})^2 \right] \left[ 2\bar{M}_W^2 - 3A_0(\bar{M}_W) - \xi_W A_0(\sqrt{\xi_W} \bar{M}_W) \right],
$$

$$
T_H^Z = \frac{\bar{M}_Z^2}{32\pi^2} \left[ \Sigma_{ZZ} - \frac{2}{v} \sqrt{h}^{44} - \left\langle \frac{\delta h_{33}}{\delta \hat{h}} \right\rangle (\sqrt{h}^{33})^2 \right] \left[ 2\bar{M}_Z^2 - 3A_0(\bar{M}_Z) - \xi A_0(\sqrt{\xi_W} \bar{M}_Z) \right]. \tag{65}
$$

The ghost terms give (again, as the photon ghost term is scaleless the contribution is identically zero):

$$T_H^{u^\pm} = \frac{\bar{M}_W^2}{8\pi^2}\left[\left\langle\frac{\delta g_{11}}{\delta\hat{h}}\right\rangle(\sqrt{g}^{11})^2 - \frac{2}{v}\sqrt{h}^{44} - \left\langle\frac{\delta h_{11}}{\delta\hat{h}}\right\rangle(\sqrt{h}^{11})^2\right]\xi_W A_0(\sqrt{\xi_W}\bar{M}_W), \quad (66)$$

$$T_H^{u^z} = \frac{\bar{M}_Z^2}{16\pi^2}\left[\Sigma_{ZZ} - \frac{2}{v}\sqrt{h}^{44} - \left\langle\frac{\delta h_{33}}{\delta\hat{h}}\right\rangle(\sqrt{h}^{33})^2\right]\xi_W A_0(\sqrt{\xi_W}\bar{M}_Z), \quad (67)$$

and for the goldstone bosons we find:

$$T_H^{\Phi^\pm} = \frac{\bar{M}_W^2}{16\pi^2}\left[\frac{2}{v}\sqrt{h}^{44} + \left\langle\frac{\delta h_{11}}{\delta\hat{h}}\right\rangle(\sqrt{h}^{11})^2 + \left\langle\frac{\delta g^{11}}{\delta\hat{h}}\right\rangle(\sqrt{g}^{11})^2\right]\xi_W A_0(\sqrt{\xi_W}\bar{M}_W) \quad (68)$$

$$+ \frac{v}{32\pi^2}(\sqrt{h}^{11})^2\sqrt{h}^{44}\left(4\lambda - \sum_{n=3}^\infty\frac{1}{2^{n-3}}\binom{n}{1,1,n-2}v^{2n-4}c_H^{(2n)}\right)A_0(\sqrt{\xi_W}\bar{M}_W),$$

$$T_H^{\Phi^0} = \frac{\bar{M}_Z^2}{32\pi^2}\left[\frac{2}{v}\sqrt{h}^{44} + \left\langle\frac{\delta h_{33}}{\delta\hat{h}}\right\rangle(\sqrt{h}^{33})^2 - \Sigma_{ZZ}\right]\xi_W A_0(\sqrt{\xi_W}\bar{M}_Z) \quad (69)$$

$$+ \frac{v}{64\pi^2}(\sqrt{h}^{33})^2\sqrt{h}^{44}\left(4\lambda - \sum_{n=3}^\infty\frac{1}{2^{n-3}}\binom{n}{1,1,n-2}v^{2n-4}c_H^{(2n)}\right)A_0(\sqrt{\xi}\bar{M}_Z).$$

Noting the raised indices in $\delta g^{11}$ for the $\Phi^\pm$ contribution, we see that the $\xi_W$ dependent parts of the $W$ and $Z$ loops are cancelled exactly by the ghost and goldstone terms, and only the $\lambda$ and $c_H^{(n)}$ gauge-parameter dependent terms remain for the scalars. This is exactly as was found for the SM Tadpole in the background field methodology [7]. Interestingly, the behavior goes beyond the SM-like interactions and also holds for the interactions which only occur in the SMEFT, i.e. those proportional to $\delta g$ and $\delta h$, as well. This also means that the $\lambda$ and $c_H^{(n)}$ terms are gauge dependent and therefore so is the tadpole. This is also consistent with [7], where they found this dependence exactly cancels against that of the Higgs two-point function and the loop contributions in the process $H\to\gamma\gamma$ at order $\frac{1}{\Lambda^2}$ in the SMEFT, leaving the observable process $H\to\gamma\gamma$ gauge invariant as it must be.

The sum of the vectors, ghosts, and goldstone bosons, neglecting $\lambda$ and $c_H^{(n)}$ dependence is given by:

$$T_H^{V,u,\Phi} = \frac{\bar{M}_W^2}{16\pi^2}\left[(\sqrt{g}^{11})^2\left\langle\frac{\delta g_{11}}{\delta\hat{h}}\right\rangle - \frac{2}{v}\sqrt{h}^{44} - \left\langle\frac{\delta h_{11}}{\delta\hat{h}}\right\rangle(\sqrt{h}^{11})^2\right]\left[2\bar{M}_W^2 - 3A_0(\bar{M}_W)\right]$$

$$+ \frac{\bar{M}_Z^2}{32\pi^2}\left[\Sigma_{ZZ} - \frac{2}{v}\sqrt{h}^{44} - \left\langle\frac{\delta h_{33}}{\delta\hat{h}}\right\rangle(\sqrt{h}^{33})^2\right]\left[2\bar{M}_Z^2 - 3A_0(\bar{M}_Z)\right]. \quad (70)$$

In order to demonstrate the compactness of this expression we expand the quantity in brackets for the $W$ contribution to $\mathcal{O}(1/\Lambda^4)$ in terms of the Wilson coefficients:

$$\left[(\sqrt{g}^{11})^2\left\langle\frac{\delta g_{11}}{\delta\hat{h}}\right\rangle - \frac{2}{v}\sqrt{h}^{44} - \left\langle\frac{\delta h_{11}}{\delta\hat{h}}\right\rangle(\sqrt{h}^{11})^2\right] = -\frac{1}{v}\left[2 + \frac{v^2}{2}\left(c_{H\square}^{(6)} - c_{HD}^{(6)} + 8c_{HW}^{(6)}\right)\right.$$

$$+ \frac{v^4}{16}\left(12c_{HD}^{(8)} - 20c_{HD,2}^{(8)} + 64c_{HW}^{(8)} + 3(c_{HD}^{(6)} - 4c_{H\square}^{(6)})^2 + 16(4c_{H\square}^{(6)} - c_{HD}^{(6)})c_{HW}^{(6)} + 128c_{HW}^{(6)}\right)\right]$$

$$+ \mathcal{O}\left(\frac{1}{\Lambda^6}\right). \quad (71)$$

In the case of the $Z$ contribution the result depends on many more operator coefficients, as well as the barred mixing angles due to the dependence in $\Sigma_{ZZ}$.

The last remaining contribution is from the quantum Higgs boson, which gives:

$$
T_H^h = \frac{1}{32\pi^2}(\sqrt{h}^{44})^2\left[\bar{M}_H^2\left\langle\frac{\delta h_{44}}{\delta\hat{h}}\right\rangle + v\sqrt{h}^{44}\left(6\lambda - \sum_{n=3}^{\infty}\frac{1}{2^{n-1}}\binom{2n}{1,2,2n-3}v^{2n-4}c_H^{(2n)}\right)\right]A_0(\bar{M}_H).
$$
(72)

The sum of all the above contributions to $T_H$ in the SM limit agrees with [7], providing a useful cross check of the result. To the extent of the authors knowledge the $1/\Lambda^2$ result does not exist in the literature in the background formalism.

With all of the contributions included we can then choose a renormalization condition related to the tadpole. Returning to Eq. 14 we obtain the coefficient of the tadpole term:

$$
t \equiv \frac{\sqrt{h}^{44}v}{16}\left[16\lambda(v_0^2 - v^2) + \sum_{n=1}^{\infty}\frac{(4+2n)v^{4+2n-1}}{2^{2+n}}c_H^{(4+2n)}\right].
$$
(73)

Choosing $t = 0$ corresponds to the proper ground state [13,14] and is the scheme we choose here. At tree level this simply reproduces the condition in Eq. 15. At one loop this corresponds to cancelling the entire tadpole contribution. Introducing $\delta t$ as a counter term, we have the renormalization condition,

$$
t = t_0 - \delta t = 0,
$$
(74)

where $t_0$ corresponds to the tree level contribution. Choosing $t = 0$ corresponds to:

$$
\begin{aligned}
\delta t ={}& -T_H\\
={}& +\sum_{\psi}\frac{N_c\bar{M}_{\psi}}{4\pi^2}\left\langle\frac{\delta\mathcal{Y}^{\psi}}{\delta\hat{h}}\right\rangle A_0(\bar{M}_{\psi})\\
& -\frac{\bar{M}_W^2}{16\pi^2}\left[(\sqrt{g}^{11})^2\left\langle\frac{\delta g_{11}}{\delta\hat{h}}\right\rangle - \frac{2}{v}\sqrt{h}^{44} - \left\langle\frac{\delta h_{11}}{\delta\hat{h}}\right\rangle(\sqrt{h}^{11})^2\right]\left[2\bar{M}_W^2 - 3A_0(\bar{M}_W)\right]\\
& -\frac{\bar{M}_Z^2}{32\pi^2}\left[\Sigma_{ZZ} - \frac{2}{v}\sqrt{h}^{44} - \left\langle\frac{\delta h_{33}}{\delta\hat{h}}\right\rangle(\sqrt{h}^{33})^2\right]\left[2\bar{M}_Z^2 - 3A_0(\bar{M}_Z)\right]\\
& -\frac{v}{32\pi^2}(\sqrt{h}^{11})^2\sqrt{h}^{44}\left(4\lambda - \sum_{n=3}^{\infty}\frac{1}{2^{n-3}}\binom{n}{1,1,n-2}v^{2n-4}c_H^{(2n)}\right)A_0(\sqrt{\xi_W}\bar{M}_W)\\
& -\frac{v}{64\pi^2}(\sqrt{h}^{33})^2\sqrt{h}^{44}\left(4\lambda - \sum_{n=3}^{\infty}\frac{1}{2^{n-3}}\binom{n}{1,1,n-2}v^{2n-4}c_H^{(2n)}\right)A_0(\sqrt{\xi}\bar{M}_Z)\\
& -\frac{A_0(\bar{M}_H)}{32\pi^2}(\sqrt{h}^{44})^2\left[\bar{M}_H^2\left\langle\frac{\delta h_{44}}{\delta\hat{h}}\right\rangle + v\sqrt{h}^{44}\left(6\lambda - \sum_{n=3}^{\infty}\frac{1}{2^{n-1}}\binom{2n}{1,2,2n-3}v^{2n-4}c_H^{(2n)}\right)\right].
\end{aligned}
$$
(75)

which depends on four barred masses (counting the barred fermion mass only once), four field-space connections plus $\Sigma_{ZZ}$, $\lambda$, and the sum over $c_H^{(n)}$. Treating the sums as a single entity gives a total dependence on eleven quantities. Conversely, the standard model result depends on four masses and $\lambda$. Expanding the tadpole result in terms of the Wilson coefficients of the SMEFT and maintaining barred mass dependence instead gives 12 parameters at dimension six and 21 at $\mathcal{O}(1/\Lambda^4)$ with 9 additional parameters at each subsequent order.[5] In this context the geoSMEFT represents a clear calculational advantage over the traditional approach to the SMEFT.

Further, as we saw in the discussion about the gauge, goldstone, and ghost terms, the compactness of the geoSMEFT expressions allows for a straightforward cancellation of terms

---

[5]The number of new parameters in $h_{IJ}$, $g_{AB}$, and $\mathcal{Y}$ at a given dimension above six stays constant, see Table 1 of [10].

which would be unclear when expanded in terms of the many Wilson coefficients contributing to each process. Similar simplifications of expressions can be expected for higher $n$-point functions, and as these expressions will generally be more complicated than those of the tadpole this simplification is crucial to an analytic understanding of the SMEFT expansion at one loop.

# 6 Conclusions

We have constructed the Feynman rules necessary for the calculation of the tadpole diagram within the framework of the geoSMEFT. In doing so we have included, for the first time, the gauge fixing of the geoSMEFT and the all-orders Feynman rules related to gauge fixing which include a single background Higgs boson and two other particles. We proceeded to calculate all diagrams contributing to the process. The results allowed us to fix the minimum of the Higgs potential at one loop and to all orders in the SMEFT power counting. In doing so we demonstrated the simplicity of expressions obtained in the geoSMEFT as compared with those expanded in terms of the Wilson coefficients which is necessary in standard approaches to the SMEFT. Further we obtained not only the first one-loop calculation including full next to leading order results in the SMEFT, but the first one-loop calculation including all orders contributions in $1/\Lambda^n$. As discussed in the introduction and Appendix B, the tadpole diagram is not only essential to fully defining one-loop results, such as the masses of the gauge bosons, but is also essential for the gauge invariance of the theory at one loop. This demonstrates the foundational nature of this work toward future precision calculations in the geoSMEFT.

Beyond the scope of the calculations contained in this article, we note that the geo-SMEFT is currently defined to include vertices of up to any three particles accompanied by arbitrarily many scalar field insertions. This has presented the opportunity for many all-orders results at tree level [6, 11] and their projection to order $1/\Lambda^4$ in phenomenological studies. This allows for the possibility to perform a truncation error analysis more consistent with the SMEFT than those commonly used where partial dimension-six squared results are used to estimate the truncation error. While few additional one-loop calculations are currently possible in the framework of the geoSMEFT, it is possible to systematically extend the geoSMEFT to include any $N$ particles plus arbitrarily many scalar field insertions. In particular, the expansion in the vacuum expectation value can be defined for arbitrary $n$–point functions by simply defining the field-space connections for ever increasing numbers of fields, i.e. for increased numbers of variations in Eq. 2. The derivative expansion is more difficult as, beyond three points functions, arbitrary powers of the momenta can be included leading to an infinite number of operators contributing to any given $n$–point function [10]. Nonetheless, the derivative expansion can separately be truncated at a given order. This will allow for the all orders in $(v/\Lambda)^n$, as well as $(p/\Lambda)^n$ to a truncated order, calculation of all two-point functions in the near future and subsequently higher $n$–point functions. With all orders results at tree- and one-loop level we can then define a fully consistent truncation error associated with the SMEFT. This is an important step toward a precision program for the studies at the High Luminosity LHC as well as for supporting and informing the case for next generation colliders.

# Acknowledgements

TC thanks M. Trott, A. Martin, and J. Talbert for useful discussions and their reading of the manuscript.

**Funding information**

TC acknowledges funding from European Union's Horizon 2020 research and innovation program under the Marie Sklodowska-Curie grant agreement No. 890787.

# A  Useful geoSMEFT definitions and relations

The following definitions and geometric relations are used extensively throughout this work in order to simplify expressions and retain them in the geometric formulation. These relations can be found in [10]. The following matrices are used to define the covariant derivatives, field strength tensors, and field-space connections:

$$
\gamma_{1,J}^{I} = \begin{bmatrix} 0 & 0 & 0 & -1 \\ 0 & 0 & -1 & 0 \\ 0 & 1 & 0 & 0 \\ 1 & 0 & 0 & 0 \end{bmatrix}, \quad
\gamma_{2,J}^{I} = \begin{bmatrix} 0 & 0 & 1 & 0 \\ 0 & 0 & 0 & -1 \\ -1 & 0 & 0 & 0 \\ 0 & -1 & 0 & 0 \end{bmatrix}, \tag{76}
$$

$$
\gamma_{3,J}^{I} = \begin{bmatrix} 0 & -1 & 0 & 0 \\ 1 & 0 & 0 & 0 \\ 0 & 0 & 0 & -1 \\ 0 & 0 & 1 & 0 \end{bmatrix}, \quad
\gamma_{4,J}^{I} = \begin{bmatrix} 0 & -1 & 0 & 0 \\ 1 & 0 & 0 & 0 \\ 0 & 0 & 0 & 1 \\ 0 & 0 & -1 & 0 \end{bmatrix}, \tag{77}
$$

as well as:

$$
\Gamma_{1,J}^{I} = \begin{bmatrix} 0 & 0 & 1 & 0 \\ 0 & 0 & 0 & -1 \\ 1 & 0 & 0 & 0 \\ 0 & -1 & 0 & 0 \end{bmatrix}, \quad
\Gamma_{2,J}^{I} = \begin{bmatrix} 0 & 0 & 0 & 1 \\ 0 & 0 & 1 & 0 \\ 0 & 1 & 0 & 0 \\ 1 & 0 & 0 & 0 \end{bmatrix}, \tag{78}
$$

$$
\Gamma_{3,J}^{I} = \begin{bmatrix} -1 & 0 & 0 & 0 \\ 0 & -1 & 0 & 0 \\ 0 & 0 & 1 & 0 \\ 0 & 0 & 0 & 1 \end{bmatrix}, \quad
\Gamma_{4,J}^{I} = \begin{bmatrix} -1 & 0 & 0 & 0 \\ 0 & -1 & 0 & 0 \\ 0 & 0 & -1 & 0 \\ 0 & 0 & 0 & -1 \end{bmatrix}. \tag{79}
$$

The quantities with tildes are defined as:

$$
\begin{aligned}
\tilde{\epsilon}^{A}{}_{BC} &= g_2 \epsilon^{A}{}_{BC}, & \text{with } \tilde{\epsilon}^{1}{}_{23} = g_2 & \quad \text{and } \tilde{\epsilon}^{4}{}_{BC} = 0, \\
\tilde{\gamma}_{A,J}^{I} &= \begin{cases} g_2 \gamma_{A,J}^{I}, & \text{for} \quad A = 1, 2, 3, \\ g_1 \gamma_{A,J}^{I}, & \text{for} \quad A = 4. \end{cases}
\end{aligned} \tag{80}
$$

The relation between barred and unbarred couplings is:

$$
\bar{g}_2 = g_2 \sqrt{g}^{11} = g_2 \sqrt{g}^{22}, \tag{81}
$$

$$
\bar{g}_Z = \frac{g_2}{c_{\theta_Z}^2} \left( \bar{c}_W \sqrt{g}^{33} - \bar{s}_W \sqrt{g}^{34} \right) = \frac{g_1}{s_{\theta_Z}^2} \left( \bar{s}_W \sqrt{g}^{44} - \bar{c}_W \sqrt{g}^{34} \right), \tag{82}
$$

$$
\bar{e} = g_1 \left( \bar{s}_W \sqrt{g}^{33} + \bar{c}_W \sqrt{g}^{34} \right) = g_1 \left( \bar{c}_W \sqrt{g}^{44} + \bar{s}_W \sqrt{g}^{34} \right). \tag{83}
$$

The above expressions make use of the barred mixing angles:

$$
s_{\theta_Z}^2 = \frac{g_1(\sqrt{g}^{44}\bar{s}_W - \sqrt{g}^{34}\bar{c}_W)}{g_2(\sqrt{g}^{33}\bar{c}_W - \sqrt{g}^{34}\bar{s}_W) + g_1(\sqrt{g}^{44}\bar{s}_W - \sqrt{g}^{34}\bar{c}_W)}, \tag{84}
$$

$$
\bar{s}_W^2 = \frac{(g_1\sqrt{g}^{44} - g_2\sqrt{g}^{34})^2}{g_1^2[(\sqrt{g}^{34})^2 + (\sqrt{g}^{44})^2] + g_2^2[(\sqrt{g}^{33})^2 + (\sqrt{g}^{34})^2] - 2g_1 g_2 \sqrt{g}^{34}(\sqrt{g}^{33} + \sqrt{g}^{44})}. \tag{85}
$$

The barred masses are given by:

$$\bar{M}_W^2 = \frac{\bar{g}_2^2}{4}\sqrt{h_{11}}^2 v^2, \tag{86}$$

$$\bar{M}_Z^2 = \frac{\bar{g}_Z^2}{4}\sqrt{h_{33}}^2 v^2, \tag{87}$$

$$\bar{M}_A^2 = 0. \tag{88}$$

Expanding the elements of the field-space connections of Eqs. 5, 6, and 16–18 become complicated very quickly, supporting the use of the geometric approach. Some examples of elements of the matrices include:

$$\sqrt{g}^{11} = 1 + c_{HW}^{(6)} v^2 + \frac{1}{2}\left[c_{HW}^{(8)} + 3(c_{HW}^{(6)})^2\right]v^4, \tag{89}$$

$$\sqrt{h}^{-44} = 1 + \frac{1}{4}\left[4c_{H\square}^{(6)} - c_{HD}^{(6)}\right]v^2 + \frac{1}{32}\left[3(c_{HD}^{(6)} - c_{H\square}^{(6)})^2 - 4c_{HD}^{(8)} - 4c_{HD,2}^{(8)}\right]v^4 + \mathcal{O}\left(\frac{1}{\Lambda^6}\right).$$

# B   Relevance of the tadpole to renormalization

Here we outline the importance of the tadpole to renormalization. We proceed by outline the renormalization procedure to arrive at the implications of the tadpole diagram in the FJ tadpole scheme, we do not employ the BFM here for simplicity. We loosely follow the notation of [15]. The fields are renormalized as follows:

$$h_0 = \sqrt{Z_{\hat{h}}}h_R, \tag{90}$$

$$W_0^\pm = \sqrt{Z_W}W_R^\pm. \tag{91}$$

The fourth component of the real scalar field is renormalized as:

$$\phi_4 = v_0 + \hat{h}_0 \to Z_v v_R + \delta v + \sqrt{Z_h}h_R. \tag{92}$$

Expanding the scalar potential of Eq. 14 about the tree level vacuum expectation value and adding the one-loop tadpole contribution we find:

$$t = -2\lambda_R v_R \delta v + T_H \equiv \delta t + T_H. \tag{93}$$

This defines the relationship between $\delta t$ and $\delta v$, in the main text $\delta t$ is chosen such that $t = 0$. This is equivalent to the choice:

$$\delta v = \frac{1}{2\lambda_R v_R^2}T_H = \frac{1}{M_{H,R}^2}T_H. \tag{94}$$

Employing an on-shell renormalization scheme as in [15] the one loop shifts in masses of the vector bosons ($V = W, Z$) are given by:

$$\frac{\bar{m}_{V,R}^2}{\bar{m}_V^2} = 1 + 2\frac{\delta v}{v} - \frac{\delta m_V^2}{m_V^2}, \tag{95}$$

where $\delta v$ corresponds to the correction outlined above, and $\delta m_V^2$ corresponds to the explicit contribution from the transverse part of the one-loop two-point functions:

$$\delta m_V^2 = \text{Re}[\Sigma_T^{VV}(M_P^2)]. \tag{96}$$

In this way we can see from Eq. 95 that even in the FJ tadpole scheme employed in this article, the one-loop tadpole is still phenomenologically relevant as it shifts the masses of the gauge bosons. Further, as the tadpole was found to be gauge-parameter dependent in Sec 5, we see that the gauge-independence of results such as the shifted masses depend on the tadpole diagram. In this way we have demonstrated the importance of the tadpole diagram to the future one-loop geoSMEFT program both phenomenologically and in terms of gauge invariance of the theory, which is necessary for the consistency of the QFT.

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
