# Peer review of "The one-loop tadpole in the geoSMEFT"

_SciPost Physics, doi:SciPost Phys. 11, 097 (2021)_

## Round 1 · Referee Report · Anonymous (Referee 1) · 2021-9-1

Strengths

  1. The work is novel and well-motivated. Potentially the first one in a series of publications using this formalism.
  2. The manuscript is well written
  3. Motivation and conclusions are convincing and stated clearly
  4. The calculation is solid and well documented.

Weaknesses

Weaknesses are minor.
The way the results are presented is not impeccable and there is a certain notation issue, that is partially inherited and that I describe in more detail in the Requested changes.

Report

The manuscript reports on the first 1-loop calculation in the framework of the geometric Standard Model Effective Field Theory ("geoSMEFT"), namely that of the tadpole diagram.

The calculation is well motivated and promising from the theoretical point of view. In addition, it elucidates some interesting aspects of the theory.

The work is solid and I definitely recommend it for publication. All requested changes are minor and mainly address the way the results are presented.

Requested changes

  1. At the end of the introduction, it is stated that, because only 3-point vertices are included, ''the geoSMEFT is only suitable for the calculation of the tadpole diagram''. The conclusions say that ''it is possible to systematically extend the geoSMEFT to include any $N$ particles plus arbitrarily many scalar field insertions''. Would it be possible to have a more quantitative statement? Are applications beyond the tadpole under development, or are there structural limitations?

  2. Symbols are often used before being defined. In some cases this makes the discussion hard to follow. For instance:

  3. The sentence ''unhatted fields are understood to be quantum fields [...] to zero'' on p9 could be moved up to the end of Sec 2, to highlight the presence of the hats (and their meaning) in Eq (19).
  4. The definition of the metric $g$ and $h$ could be moved up (starting from the Lagrangian rather than from the field rotations)
  5. The definition of $\Pi_{1,2}$ in Eq (22) could be moved to before Eq (20) and simplify already the expression there.

  6. It would be nice to have explicitly the relations between bosons in mass and gauge eigenbases through the $\mathcal{U}$ and $\mathcal{V}$ matrices (the analogous of Eq (6))

  7. Before Eq (6), I would remove the reference to the FeynRules output format and just state the conventions, as for instance it got me wondering whether $p_1$ in ${\hat h, G_1, G_2}$ was the momentum of $\hat h$ or $G_1$.

  8. Could Eqs (33)-(37) be written in a more compact way?

  9. in Eq (16) : what is $\stackrel{(\sim)}{H}(\phi_I)[Y_\psi]$? And $\stackrel{(\sim)}{H}(\psi_I)$?

  10. in Eq (19), why is the fermion field not split into quantum and background components?

  11. As a general comment, I would suggest investing in notation adjustments, if not for his work, at least for future geoSMEFT publications. The proliferation of new objects is undoubtedly problematic and further worsened here by the use of the BFM, but the result is a quite thick forest of new symbols with many indices and decorations, where the reader can have a hard time finding his/her way. Sometimes the same symbols indicate different things and there are groups of symbols that differ only by the latin/mathcal/greek font, by upper/lowercase or by the addition of a $\sim$, bar or hat on top, but denote very different objects. As far as I can tell, there is no meaning associated to these choices (e.g."all mass eigenbasis objects are in mathcal") so one always has to go back and forth to check the definitions one by one.

Some specific examples: - $\kappa$ represents different objects depending on whether it has $IJ$, $\mathcal{AB}$ or $\mathcal{ABC}$ indices or no indices at all (Eq (7)). The same goes for $f$ (Eq (7) vs (10)). In Eqs (20), (21), (51), (59) why isn't $(\sqrt{\kappa^{-1}})^2=\kappa^{-1}$? Is $\kappa$ a matrix or a scalar quantity here? - $h$ is used both for a metric and for the Higgs field. The latter appears after Eq (5) but without explicit mention of what it represents. - are $g_{AB}$ and $\hat g_{AB}$ the same? $\hat g$ appears in Eq (48) but I do not see it defined. The last line of page 9 hints at it having an expectation value along the same lines as $\hat h$ and $\hat\phi$, which is somewhat mysterious. - $A$ indicates sometimes the index running over $0\dots 4$, sometimes the photon (e.g. $\bar u^A$ in eq (61) is exactly the same notation as in (58), but the upper index is something else). - Some of the objects defined implicitly depend on fields and Wilson coefficients, but this is not stated clearly at the beginning and it's hard to guess for a non-expert. It would be nice if this was indicated at least once, when they are first introduced. - In sec 2 $\mathcal{G}$ are gluons (I'm not sure if in the mass or gauge basis). In Eq. (50) the gluons become $G$, while $\mathcal{G}$ is now the gauge fixing term of SU(2).

Is there anything that could be done to help the reader? Maybe adding a table? On top of this, I discovered as late as at p12 that raised and lowered indices in the metrics are understood with GR conventions and therefore differ by a sign. This should at least be specified in Sec 2, together with an indication of the sign convention.

  1. some typos I noticed:
  2. p2. couting $\to$ counting
  3. Eq (7), the gluon kinetic term needs either to remove the indices from $\kappa$ or to change one of the two $\mathcal{A}$ in $\mathcal{B}$. In the $\bar \psi\psi$ term $\mathcal{Y}$ should be $\mathcal{Y}^\psi$. In the chromomagnetic term, $T_A\to T_{\mathcal{A}}$. In the $L_{IA}$ term, there is an index 1 missing on the first fermion. In the $G^3$ terms Lorentz indices are in a funny order, that results in a minus sign wrt the $W^3$ term. Is this wanted?
  4. Eq (18) there is a spurious bracket ]
  5. end of Sec 2. in "the couplings $\hat h$ to two quantum fields'' an "of'' is missing
  6. in Eq (20) $\mu\nu$ should be $\mu_1\mu_2$
  7. in Eq (24) the flavor indices go from $rr$ to $pp$ between the first and second line.
  8. in Eq (29), (30) should the $i,j$ indices be capital $I,J$? Or even $A,B$, for consistency.
  9. One between $G^{\mathcal{A}}_{\rm color}$ in Eq (48) and $\mathcal{G}^{\mathcal{A}}_{\rm color}$ in Eq (50) should be adjusted to match the other.
  10. in Eq (52) - (54), (60), (62) should $\xi$ be $\xi_W$?
  11. in Eq (64) are the $c_{\psi H}$ coefficients missing in the sum?
  12. before Eq (73). depedence $\to$ dependence
  13. end of p12. "the the'' $\to$ "the''
  14. Eq. (92) is a $+\mathcal{O}\left(\frac{1}{\Lambda^6}\right)$ missing?

---

## Round 2 · Referee Report · Anonymous (Referee 2) · 2021-10-17

Report

I agree with Referee 1, who has provided detailed comments on the manuscript. The author has responded to most of the changes requested. However, as also implied by Referee 1, it may be helpful to reader if more explanations of the background of the subject is provided before presenting the technical results, to increase the readability of the paper.

A minor grammatical point: "where" is not usually used to start a sentence, but a subordinate clause.

---

## Round 2 · Referee Report · Anonymous (Referee 1) · 2021-10-18

Report

The author has replied satisfactorily to all the points raised. The present version is fully suitable for publication.

---

## Round 2 · Author Response

We thank the referee for their support of the manuscript as well as their thorough notes. We have attempted to implement them all. In the case of notational suggestions we have done our best to balance the referee's suggestions with maintaining the original notation introduced in refs 10 and 11. It is clear from the referee report that as the geoSMEFT is developed further more effort needs to be put into clarifying the notation for readers not familiar with the development of the geoSMEFT. In what follows all mentioned equation numbers correspond to the equation numbers of the updated manuscript.

---

## Round 2 · List of Changes

1) We have rephrased this discussion in the conclusions to indicate what parts are well defined and which need further development. The plan is to continue to extend the one-loop geoSMEFT calculations in the near future.

2) a) added discussion of hatted fields near Eq19 b) we have moved the introduction of "h" and "g" as suggested c) we have introduced Pi_12 earlier in the text as suggested

3) We have included such an expression in the new Eq 8.

4) We have followed the referee's suggestion and clarified the notation of the rules.

5) We have used the antisymmetry of this field-space connection to reduce the number of terms in the expressions. Unfortunately there does not appear to be a way to reduce the first two lines without introducing a transformation matrix that takes $\kappa^1\leftrightarrow\kappa^2$ which would be more cumbersome than the current expressions.

6) We have clarified the notation below (new) Eq. 18 and corrected the mistake H(psi)->H(phi)

7) We have added "As fermionic fields are not involved in the gauge fixing they are not split into background and quantum fields." to the text at the end of Section 2.

8) We agree with the referee that the notation can be cumbersome. This article sought to be a balance between the original notation developed in citations 10 and 11 and a clearer picture of the geometry as the geoSMEFT has evolved over the last year. We hope to further clarify this notation as we continue to develop the one-loop program.

a) We have added the comment: "Latin indices $A,B,\cdots$ are those associated with the four-component representation of the gauge boson indices for $SU(2)_L\times U(1)_Y$, $I,J,\cdots$ are are the four-component indices associated with the four-component real scalar field, and $\mathcal A,\mathcal B$ are associated with color indices are of the gluons." below (new) Eq20.

We have dropped the use of $\kappa_\mathcal A\mathcal B$ in favor of the scalar quantity $\kappa$ for clarity. We feel that changing the naming convention of the field-space connections that share $\kappa$ would be too far a change from the original authors' work for this short note, but is something to consider in future development of the geoSMEFT.

b) We have added a comment to distinguish the h field from the h metric below (new) Eq.7

c) The hatted field space connections were defined below (new) Eq 49, we have clarified the language by adding "(i.e. $\hat g$ and $\hat h$)" to further clarify. \hat g has an expectation value, as do all field-space connections. This simply corresponds to setting the field dependence in the field-space connection to zero, it does not correspond to an expectation of a gauge boson field. We hope this clarifies what the referee found mysterious.

d) We have changed the notation for the ghost field to $u^\gamma$ to clarify and added a note to the text.

e) There is no implicit field dependence, however the explicit field dependence has implicit dependence on the rotations $\mathcal U$ and $\mathcal V$. This is noted below Eq24. We have added an extra comment on this below (new) Eq. 6. "As the scalar field $\phi$ is related to its mass eigenstate field $\Phi$ by the inverse square roots of the expectations of these matrices, they are (in the mass eigenstate basis) implicitly dependent on $\sqrt{h}$. "

f) $G$ are the canonically defined gluon fields, the gauge fixing term has been corrected to be $\mathcal G$. Below Eq 9 we have also added "$G$ corresponds to the canonically normalized gluonic field, while $\mathcal G$ corresponds to the gluonic field before the kinetic term is transformed."

g) Footnote 1 was added to clarify the difference between the raised and lowered indices in the field-space connections. This does not correspond to a simple sign flip - the off diagonal components of the field-space connections result in nontrivial relationships between the matrices that were solved for perturbatively in this manuscript.

9) We corrected all typos pointed out by the referee.

---

## Editorial Decision

published